# Application of Nanotechnology in Plant Genetic Engineering

**DOI:** 10.3390/ijms241914836

**Published:** 2023-10-02

**Authors:** Kexin Wu, Changbin Xu, Tong Li, Haijie Ma, Jinli Gong, Xiaolong Li, Xuepeng Sun, Xiaoli Hu

**Affiliations:** 1Collaborative Innovation Center for Efficient and Green Production of Agriculture in Mountainous Areas of Zhejiang Province, College of Horticulture Science, Zhejiang A&F University, Hangzhou 311300, China; 2Key Laboratory of Quality and Safety Control for Subtropical Fruit and Vegetable, Ministry of Agriculture and Rural Affairs, Hangzhou 311300, China

**Keywords:** genome engineering, plant transformation, nanotechnology, nanoparticles, CRISPR

## Abstract

The ever-increasing food requirement with globally growing population demands advanced agricultural practices to improve grain yield, to gain crop resilience under unpredictable extreme weather, and to reduce production loss caused by insects and pathogens. To fulfill such requests, genome engineering technology has been applied to various plant species. To date, several generations of genome engineering methods have been developed. Among these methods, the new mainstream technology is clustered regularly interspaced short palindromic repeats (CRISPR) with nucleases. One of the most important processes in genome engineering is to deliver gene cassettes into plant cells. Conventionally used systems have several shortcomings, such as being labor- and time-consuming procedures, potential tissue damage, and low transformation efficiency. Taking advantage of nanotechnology, the nanoparticle-mediated gene delivery method presents technical superiority over conventional approaches due to its high efficiency and adaptability in different plant species. In this review, we summarize the evolution of plant biomolecular delivery methods and discussed their characteristics as well as limitations. We focused on the cutting-edge nanotechnology-based delivery system, and reviewed different types of nanoparticles, preparation of nanomaterials, mechanism of nanoparticle transport, and advanced application in plant genome engineering. On the basis of established methods, we concluded that the combination of genome editing, nanoparticle-mediated gene transformation and de novo regeneration technologies can accelerate crop improvement efficiently in the future.

## 1. Introduction

Currently, population growth, climate change, and the COVID-19 pandemic are placing enormous pressure on agriculture and food security [1,2]. To meet food demands by the global population, advancements in biotechnology are urgently needed to not only improve the crop yield and quality, but also reduce the loss caused by biotic/abiotic stresses. In recent years, genome editing plays a critical role in increasing crop stress resistance and to ensure food production and security [3]. For instance, an increase in drought stress tolerance was observed in rice plant after *OsERA1* was modified using clustered regularly interspaced short palindromic repeats (CRISPR/Cas9) [4]. In comparison to wild-type crops, the development of *SlHSA1* genome editing mutants of rice showed greater sensitivity to temperature stress [5]. Despite the rapid development of the genetic engineering method for different plants, copious plant species remain difficult to be genetically transformed. This is largely because current research has not invented a passive, efficient and species-independent way to bypass the obstruction of the cell wall [6,7]. Amongst predominant plant genetic engineering methods, Agrobacterium-mediated transformation is a widely used tool but subject to a limited variety of plants. Gene-gun (biolistic particle) transformation is a plant species-independent delivery method yet can result in tissue damage and inefficient integration efficiency [8]. Other biomolecule delivery methods such as electroporation, polyethylene glycol (PEG), the pollen tube pathway, and cationic delivery also have disadvantages, and these methods remain challenged by low efficiency, low regeneration, and cell damage and cytotoxicity [9,10].

Nanotechnology has been applied in agriculture such as nanosensors [11], nanopesticides [12], and nanofertilizers [13], and this has led to considerable interest in recent studies. Indeed, nanoparticle-mediated gene transformation has been extensively used for gene delivery in plants. As such, nanoparticles—particles of small sizes (1–100 nm) with tunable physical and chemical properties [14]—could be served as an effective vector to transport cargos to bypass the cell wall and the plasma membrane. In particular, the transit of molecular biology cargos such as DNA, RNA, and proteins to plant cells has become increasingly imperative [7,14]. Hence, nanoparticle-mediated biomolecule delivery would be a potential approach to overcome the drawbacks of conventional biomolecule delivery methods, improving transformation efficiencies in agricultural plant biotechnology. Here, we review the different methods used for plant transgenics, and emphasized the advantages in application of nanotechnology from the development of nanocarriers for plant genetic material delivery to nano-mediated plant regeneration. We also highlight the advance of nanotechnology in gene editing, and discuss the safety issues and potential threats for nanotechnology manipulation on plant transformation in the future.

## 2. Conventional Genetic Transformation Methods

Plant genetic transformation has been a popular research field in recent decades. With the advancement of conventional genetic-engineering approaches, plant biology studies have been brought into a new era. However, these genetic engineering tools are hindered by a rigid multilayered cell wall in most species, challenges in plant regeneration and their own downsides. An overview of plant transformation methods was summarized in this section.

### 2.1. Gene Gun-Mediated Transformation

Gene gun-mediated transformation (particle bombardment) is based on a biolistic particle delivery method, and is commonly adopted in genetic transformation in plants [15]. In brief, compressed gas is used to generate a cold gas shock wave into the bombardment chamber, and hit the fine gold powder carrying DNA molecules. With the gas pressure, the gold particles carrying DNA pass through the cell wall, the cell membrane, cytoplasm and other layers of structure to reach the nucleus, completing gene transfer. Gene-gun transformation successfully addresses the species-dependent restriction. Furthermore, the gene gun can be leveraged for plasmid biological delivery, which agrobacterium transformation method fails to do so. However, it is limited by plant tissue damage, low-level and sporadic expression, multiple copies, and low integration efficiency during the bombardment. Moreover, the requirements for specialized facilities and expensive materials restrict its extensive usage [16]. Otherwise, biolistic delivery would be used more frequently when targeting the nuclear genome taking advantages of the nonspecific localization of molecular cargoes [6]

### 2.2. Agrobacterium-Mediated Transformation

Agrobacterium, a Gram-negative bacterium, ubiquitous in soil that chemotaxis most dicotyledons under natural conditions and induces crown galls or hairy roots [17]. The cells of *Agrobacterium tumefaciens* and *Agrobacterium rhizogenes* contain Ti and Ri plasmids, respectively, on which there is a section of T-DNA. After Agrobacterium introduces the target gene into plant cells through infection of wounds, T-DNA can be integrated into plant genome in the form of a single strand, leading to stable transformation. Due to its high stability, simple operation, and high efficiency, Agrobacterium-mediated transformation has been the most widely used technique in plant genetic engineering. It can be roughly divided into in vitro systems and in vivo/in planta systems. The in vitro systems require the use of different types of sterile explants, such as leaves [18], petioles [19], hypocotyls [20], stem internodes [21], stem segments [22], roots [23], cell suspension cultures [24], and calli [25]. The processes are trivial and time-consuming as the regeneration is obtained from a single transformed cell. Whereas the in planta system can be applied to reproductive organs (such as pollen tube), axillary meristem, and others in situ. Since the pollen tube pathway was reported in 1983, this method has been widely used by scientists [26]. After pollination, the ovary is injected with the DNA solution containing the target gene, and the pollen tube channels are used to introduce and further integrate the exogenous DNA into the genome of recipient cells. With the development of the fertilized egg, it becomes a new individual with transgenes. 

Recently, the Cas9/sgRNAs has been successfully leveraged for editing target genes via distributing plasmids into plant cells through foliar spraying [27]. Although it is simple, rapid, and efficient, the tissue culture-independent pollen tube pathway can only be used in flowering plants, in particular for Arabidopsis, and happens to cause unstable transformation [28]. Therefore, de novo meristem regeneration has been developed. Gene-edited dicotyledonous plants have been generated through this method [29], which successfully bypasses the need for in vitro tissue culture.

Nevertheless, the Agrobacterium-based system has two main limitations. On the one hand, it is amenable only to DNA delivery and suffers from host limitation [30,31,32]. On the other hand, researchers have found that Agrobacterium-mediated delivery may result in undesired traits caused by random DNA integration and endogenous plant gene disruption [32,33]. Moreover, the use of Agrobacterium for biomolecule delivery raises public and regulatory concern as it yields genetically modified (GM) organisms. For instance, plants transformed by Agrobacterium are under the regulation of the US Department of Agriculture (USDA) [34]. In Europe, few EU countries cultivated GM crops till 2018 [35].

### 2.3. Electroporation

Electroporation could instantly improve the permeability of the cell membrane under the high-intensity electric field [36]. Under such circumstances, it allows molecules from the outside to diffuse into the cell. The technology can introduce nucleotides, DNA, RNA, proteins, dyes and virus particles into prokaryotic and eukaryotic cells. Owing to its high efficiency, controlled dosage and wider applicability [37,38], electroporation has been a versatile tool adopted in medicine and food biotechnology [39]. The applied field of the electroporation has also been expanded from in vitro research to in vivo research for intracellular delivery [37]. Gene transfer by electroporation was widely carried out in animals, yet the application of electroporation in plants was hampered by thick cell walls. Further, electroporation has the adverse effects of causing target tissue damage and improper cell function [33].

### 2.4. PEG Delivery System

In the presence of divalent cations (Mg^2+^, Ca^2+^, etc.), PEG can effectively induce DNA to form granular precipitation, which enables the cell membrane to absorb such DNA particles through endophagocytosis. PEG-mediated transformation has been a commonly used method for plant protoplast transformation [40]. The popularity of the PEG delivery system lies in the following advantages: conversion efficiency, easy operation, low cost and mild reaction conditions. However, the PEG delivery system is limited by genotype and must be applied to naked protoplasts which are commonly used for transient gene expression [41].

## 3. Nanoparticle-Mediated Biomolecule Delivery

A nanoparticle is an artificially made tiny particle under 100 nm [42], which is smaller than the cell size exclusion limits, and has tunable physical and chemical properties that allows it to be precisely controlled to interact with biological molecules. To date, nanoparticles have emerged as customizable and potential tools that are readily accessible in biomedical and agriculture [43], which has gained special interest in utilizing nanoparticles as transfer vehicles for the delivery of bioengineering molecules in both animals and plants. In particular for mammalian systems, engineered nanoparticles can harness molecular cargoes and therapeutic agents on a subcellular level, giving rise to targeted delivery and precise control of cargo release to the nucleus or organelles [6]. Furthermore, knowledge acquired from biomolecule delivery to animals guides the transformation system in plants, and could expedite advancements in crop breeding and crop resilience. Nanoparticles have the potential to enter cells without external assistance compared to traditional gene transformation methods, which has been demonstrated in several studies [16].

### 3.1. Type of Nanoparticles 

Various types of nanoparticles (NPs) include DNA nanostructures, peptide nanoparticles, silicon-based nanomaterials, clay nanosheets, polymer-based nanomaterials and metal-based NPs, which have been elaborated elsewhere [44,45]. Each type of NPs has its own characteristics and functions, leading to transient or stable transformation. Many NPs have been used for plant transformation. For example, the magnetic NP was loaded with DNA and transferred into pollen to directly generate stable transgenic seeds without regeneration [46]. However, the efficiency of this method was still in controversy [47]. In 2022, another study was reported to successfully prove that the nanomagnetic beads could introduce foreign genes into maize pollen through the pollen germination pore [48]. Carbon nanotubes (including single-walled carbon nanotubes and multiwalled carbon nanotubes) show good biocompatibility. Demirer et al. [49] demonstrated that a large number of plasmid DNA can be loaded by enhancing the aspect ratio of carbon nanotubes, and carbon nanotubes can penetrate cell walls to achieve transient transformation. In addition, some studies have shown that carbon nanotubes deliver small RNA to intact plant cells by electrostatic adsorption and can protect them from degradation [49,50]. Most of these NPs guide transient transformation and do not integrate exogenous genes into the host chromosome. Hence, NP-mediated gene transformation has the advantages of easy operation, a short cycle, higher expression efficiency and biosafety due to not producing heritable offspring. Furthermore, this feature of NPs may bypass genome modification regulation and alleviate the public concerns about the food safety of transgenic crops [9,51].

### 3.2. Methods for Nanomaterial Preparation

The nanomaterial can be synthesized physically, chemically or biologically using an approach called top-down or bottom-up synthesis [52] (Figure 1). For the top-down approach, a physical process is usually used to divide the bulk materials into small molecules, which are further converted into NPs as needed. In the bottom-up approach, nanomaterials can be prepared directly through the construction of small nanostructures, which is generally achieved by chemical methods. Recently, green synthetic methods using plant extracts or other microorganisms to produce NPs have emerged as a promising tool. At present, common heavy metal ions including Ag, Cu, Co, and Ni have been used to synthesize mental oxide NPs [53,54]. The generation of NPs from plant extracts avoids high costs and reduces toxicity compared to traditional physical and chemical methods [55]. In practice, a large body of literature emphasizes that NP size, shape, and physical, chemical, and biological characteristics directly influence the effectiveness of transportation [56,57,58], thus it is of great significance to understand the design and synthesis method to adjust the properties of NPs [59]. The novel synthesis method may serve as the optimal platform to construct NPs suitable for intracellular biomolecule delivery because of its high efficiency and simplicity and because it is pollution free.

### 3.3. Transportation of NPs in Plants

Uptake and transport of nanoparticles in plants are limited by various factors such as plant species, particle concentrations, particle size, surface charge, and exposure time [60]. The absorption of NPs in plants is mainly through leaves and roots (Figure 1). Meanwhile, their different morphological characteristics lead to different transport barriers for NPs [45,61]. The epidermis of plant leaves is often covered with a defensive cuticle, which consists of an insoluble epidermal membrane impregnated and covered with soluble wax. Therefore, NPs larger than 10 nm enter cells through stomata. In addition, stomata number and activity vary among species and depend on environmental conditions, so absorption of NPs needs to take these conditions into account [62]. For instance, the effect of stomatal opening or closing on the absorption of ZnO NPs was investigated in wheat leaves. The results showed that the concentration of detectable zinc in chloroplasts and cytoplasm decreased by 33.2% and 8.3%, respectively, with a decrease in stomatal diameter [63]. When absorbed by roots, it is not only epidermal cells in the root that restrict large NPs, but the Casparian strip in the endodermis also limits the transport of NPs. NPs may only cross the endodermis through damaged roots, secondary root birth sites, and passage cells. Some studies have also used injection or petiole feeding to break through the cuticle of leaves to deliver the NPs suspension into the plant cells purposely [61]. Although these are effective ways to channel foreign matter into cells, they have the disadvantage of being low throughput and laborious, and therefore have not been widely adopted [45]. A more commonly used method is injection-free penetration. Alternatively, current research has proven that some NPs can drill holes in the cuticle themselves [53].

After passing through the cuticle, NPs need to cross the mesophyll cells to reach the vascular system, which is necessary for NPs to be transported over long distances in plants [64]. The internalization of NPs in plant mesophyll cells is mainly limited by the cell wall, and secondarily by the cell membrane [3]. The methods for nanomaterials to penetrate the cell wall and membranes included diffusion, endocytosis, the plasmodesmata or by physical and chemical destruction [7]. By changing the composition of the NP osmotic buffer, the local water potential of the cell wall can be changed to control the strength and tension of the cell wall, so as to achieve the internalization of NPs [45]. Some DNA-coated NPs, such as gold microparticles, are still transported through tissue into cells depend on biomolecule delivery methods such as a gene gun [65,66]. Furthermore, the interactions between NPs and various membranes within plant cells also need to be fundamentally understood so that the NPs can be transported without external assistance [67]. Several mechanisms for nanoparticle transport through the lipid bilayer, such as passive penetration and endocytosis, have been proposed by some studies, but many uncertainties remain regarding these mechanisms, especially on recent proposals that efficient cargo transport is possible without internalization of nanoparticle cells [14,68]. Further, the shape and charge status of nanoparticles can also influence their entry into plant mesophyll tissues. Rod-shaped gold nanoparticles were more easily absorbed and internalized by Arabidopsis leaves than spherical nanoparticles at similar particle sizes [14]. Electrostatic attraction should also be considered, as the adsorption of positively charged nanoparticles was stronger than that of negatively charged nanoparticles when plant cell walls are mainly negatively charged [69].

The transport of NPs in mesophyll cells is mainly through the symplastic and apoplastic pathways. The apoplast is a system of cell walls, intercellular spaces and vessels around the protoplast of plant cells. Conversely, a symplast is a totality of interrelated protoplasm connected by plasmodesmata that pass through the cell wall. In general, fluids encounter less resistance when moving through the apoplastic pathways. Recent findings suggest that fluid flow in the apoplast will be restricted by the plant after the introduction of NPs, and it has been speculated that the apoplast may be the translocation pathway of NPs in plants. However, the main translocation pathway of NPs has not been determined, and more mechanisms are urgently needed to quantify the interaction between NPs and the plant cell wall [64]. Combining knowledge from multiple disciplines and applying a multiscale approach may bypass these debates.

Currently, it is necessary to further study the specific transport channel of NPs and corresponding mechanisms, which is important for guiding NPs to reach the desired destination [61]. Some studies advance mathematical models to study the mechanism of NP cell transport [65]. It has been demonstrated that NP aspect ratio and stiffness may be key factors during nanomaterial transport across cell walls [53]. The physical and chemical properties of nanomaterials likewise affect NP transport. In addition, morphological and physiological differences among plant species also influence the uptake and translocation of NPs [70]. Elucidating the mechanisms of NP transport within living plant cells is therefore of significant importance for improving plant genetic engineering applications and NP-based agri-technologies [65]. 

## 4. Application

### 4.1. Application of Nanotechnology in Plant Genetic Transformation

NP-mediated gene transformation systems have been successfully applied by using leaf, root or protoplasts of model plants and crops to promote transient and stable transformation efficiency (Table 1). The small size, biocompatible, tunable properties and variability of nanomaterials make them promising biomolecular carriers in plants. The most important role of nanotechnology for plant genetic engineering is the ability to control the delivery of goods to different plant species and tissues. Some nanoparticles can also image the delivery and release of goods due to their inherent fluorescence properties. During transportation, studies have also shown that NPs endow cargoes with resistance against degradation [68]. Hence, NPs can be leveraged not only to deliver plasmid DNA into intact plant cells for genetic transformation, but also can delivery small interfering RNA to achieve post-transcriptional gene silencing, conferring disease resistance in crops [32]. Zhang et al. [31] have successfully used DNA nanostructures and carbon nanotubes to deliver sRNA directly in realizing effective gene knockout [31,71]. Meanwhile, it is possible to distribute proteins to plants using NPs, which will make DNA-free editing possible and further facilitate the development of genome editing in plants [7].

NP-mediated plant gene transformation targets involve mature plants and plant suspension cells. Although mature plants are more predominantly transformed, it has been reported that transformed suspended plant cells may have greater biomedical advantages. The transformation of mature plants can only result in small-area transformation by common injection, while the transformation of suspended cells is purer and more efficient. Moreover, suspension cells are more likely to allow NPs to enter the cell [68]. Conventional plant gene transformation involves introducing DNA into the nucleus to produce transgenic plants, while genetic engineering for organelles (such as chloroplasts and mitochondria) can be inherited maternally without the restriction of transgenes [9]. The functionalization of NPs with biomolecules also enables gene transfer to the nucleus and chloroplast [87,88]. Carbon nanotubes, for example, can be chemically modified to deliver DNA to organelles in a variety of plants. These studies can be achieved without transgenes and are applicable to both model and non-model plants [49]. Accordingly, the transport mechanism of nanoparticles into subcellular organelles through the plasma membrane still needs to be further explored [65]. In order to mitigate the influence of the cell wall on gene transformation, protoplasts have also been used for gene transfer. However, it is still challenging to effectively culture and regenerate plants from protoplasts in most species [89].

### 4.2. Application of Nanotechnology in Gene Editing

Genome editing is a newly developed genetic engineering technology that can modify specific target genes in the genome of an organism [90]. The early genome engineering technique can only randomly insert foreign or endogenous genetic material into the host genome. However, the emerging nuclease-enabled genome editing method creates more accurate modification of specific target genes in the genome. Today, there are four classes of engineered nucleases being used for creating specific-site double-strand breaks (DSBs), including Meganuclease [91], Zinc Finger Nucleases (ZFNs) [92], Transcription Activator-Like Effector Nucleases (TALENs) [93], and commonly used Clustered Regularly Interspaced Short Palindrome Repeats (CRISPR/Cas9) [94,95]. These methods generate site-specific DSBs at specific locations in the genome, inducing organisms to repair DSBs by non-homologous end linking (NHEJ) or homologous recombination (HR), which leads to gene replacement [96,97]. For the CRISPR/Cas9 system, two components need to be delivered simultaneously: Cas9 endonuclease and sgRNA. In genetic engineering, CRISPR plasmids can avoid the random integration of transgenes with the help of carbon nanotubes that can achieve transient expression for permanent editing [16]. Presently, three types of cargoes were adopted in the delivery of the CRISPR/Cas system. The most straightforward approach is RNP, which has no use for transcription or translation but is hampered by the high molecular weight of Cas9 protein [98]. The second delivery strategy is to transfect sgRNA plasmids. Using this method, corresponding plasmids are integrated into the same transfer plasmids to carry out gene mutations. However, the transfection efficiency will be reduced due to the large gene fragment of the CRISPR/Cas9 system. Another cargo is Cas9 mRNA. The simultaneous transfer of Cas9 mRNA and sgRNA into cells can be directly translated into proteins, yet the instability and transient expression of mRNA limit its wider application [99].

Even though the cell wall greatly impedes the transfer of genome editing cargo to mature plants, NPs have become a potential delivery platform for plant genome editing and account for its ability to target loading and transport plasmid DNA, mRNA and RNPs. Direct cytoplasmic delivery of the Cas9 protein with the sgRNA complex has been studied to provide more efficient gene editing [100]. It has been reported that lipid-based NPs, CRISPR-Gold, DNA nanoclews, and polymer nanoparticles have been used to in the CRISPR/Cas system [99]. Despite the general utility of CRISPR-Cas technologies, mono-sgRNA gene editing limits the efficiency and generation of numerous mutants at once. Instead, multiplexed CRISPR technologies have developed for multilocus editing [101]. It worth noting that some studies reported a nano-biomimetic transformation system for gene editing. The system, which can be mediated by NPs, can allow the stable transformation of crop varieties and make vectors free of residue [65,102]. The combination of nanotechnology and gene editing can highly promote crop breeding.

## 5. Conclusions

After discussing the current challenges and opportunities of transgenic plant regeneration, integrating genome editing, nanotechnology and de novo regeneration may provide breakthrough innovation in plant genetic engineering [9] (Figure 2). Much work needs to be performed to delicately design and use nanotechnology to advance gene editing and crop breeding. Although NPs may be likely to achieve unprecedented levels of accurate control at the subcellular level, the pathway still has many shortcomings. First, most NP applications still require transformation tools, such as gene guns or an electromagnetic field [49], and nanomaterial-mediated delivery is still less efficient than biological delivery methods. Thus, enhancing the delivery efficiency of CRISPR reagents is an urgent need to enable genome editing for practical applications. However, these genome editors often generate unwanted off-target editing, which is a side effect with safety concerns [103]. Second, more optimization of nanostructures is needed to bypass the plant cells. The mechanism of absorption, transport and accumulation of nanoparticles is still unclear, and the interaction between nanoparticles and plant cells, as well as the transformation of nanoparticles targeting protoplasts and suspended cells, also need to be explored from many aspects. Third, most NP-mediated transformation is transient, and it is necessary to optimize progeny regeneration for solid stable transformation to enhance the stable transformation capacity of nanoparticles in all kinds of plant species. 

## 6. Prospective

In order to solve these puzzles, it is necessary to combine multiple disciplines and study the relationship between nanomaterials and plant cells from various aspects. Application of machine learning including supervised, semi-supervised and unsupervised methods assists in the analysis of genetic and genomic datasets [105], which provides general guidelines to better predict optimized pathways and outcomes of nanotechnology application in plant biology. Additionally, convolutional neural networks, deep belief networks, multivariate Poisson deep learning, multilayer perceptron, probabilistic neural networks or radial basis function neural networks may help improve the prediction of target genes for nanotechnology-guided genetic engineer approaches by integrating heterogeneous datasets while side stepping the curse of dimensionality [106]. As many species callus can regenerate into intact plants, callus may build a bridge for the transfer of NP, and thereby overcome the limitations of the genetic transformation of species. On the other hand, threats of biosafety and plant toxicity posed by nanotechnology should also be taken into account. As genome editing may produce off-target sites, it is necessary to characterize and understand the outcomes of such side effects. Much more effort should be made to evaluate whether these side effects are harmful or acceptable [103]. For long-term assessment, the European Directive previously introduced the obligation to implement a monitoring plan to trace and identify any direct or indirect effects on humans or animals and the environment of GM organisms. Ten years of post-market environmental monitoring of GM maize was accomplished [107]. It is also crucial to continually evaluate the environmental and health risks associated with nanomaterials used in agriculture. Natural or artificial nanoparticles can be found in huge quantities in nature, and the detrimental effects of nanoparticles have been reviewed [108], thus we should not ignore the serious consequences any more. Future advancements in nanomaterials as biomolecular delivery carriers with reduced side effects will significantly enhance plant biotechnology, thus addressing various challenges encountered during grain production and predictive breeding. An innovative approach that integrates nanotechnology-enhanced genetic engineering with genomic selection which has successfully been used for genomic prediction of gene bank wheat landraces [109] can facilitate the application of nanotechnology in cultivated gene pools in the near future.

## Figures and Tables

**Figure 1 ijms-24-14836-f001:**
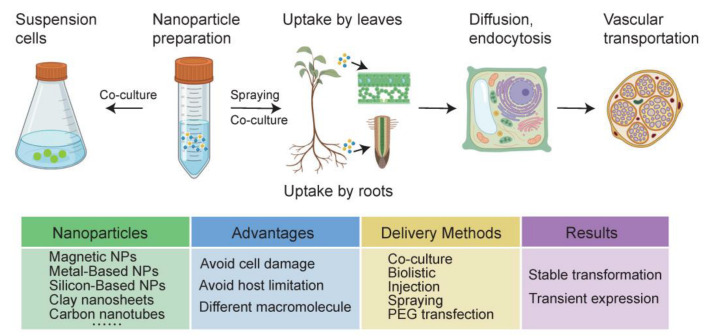
Characterization and the general process of nanoparticles (NPs) preparation and delivery for plant transformation. There are many different kinds of nanoparticles that can be applied. Nanoparticles can be leveraged to deliver DNA, RNA or protein into intact plant cells by diffusion, endocytosis, the plasmodesmata or by physical and chemical destruction finally achieving instantaneous transformation or stable transformation. NP-mediated plant gene transformation targets involve mature plants and plant suspension cells. The absorption of NPs in mature plants is mainly through leaves and roots, then transporting through the vascular system. NP-mediated plant gene transformation shows many advantages over conventional methods.

**Figure 2 ijms-24-14836-f002:**
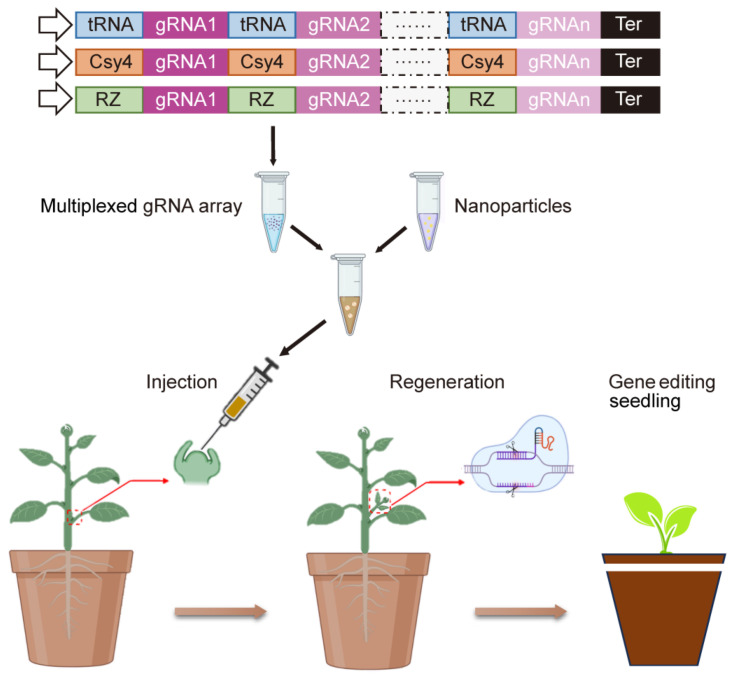
Schematic representation of a promising plant transformation method integrating genome editing, nanotechnology and de novo regeneration. Assembly of multiple guide RNAs (gRNAs) into a CRISPR vector, then transferred into intact plant meristem, resulting in tissue culture-independent transgenic plants. Adapted from [9,104].

**Table 1 ijms-24-14836-t001:** Application of nanoparticle-mediated transformation in various plant species.

Target Plant Species and Tissues	Nanoparticle Type	Cargo	Delivery Method	Stable Transformation/ Transient Expression	References
*N. benthamiana* (leaves)	Layered dihydroxide clay nanosheets	Double-stranded RNA	Spraying	Transient expression	[72]
*N. benthamiana*	Carbon nanotube	GFP plasmid	Injection	Transient expression	[49]
*E. sativa*	Carbon nanotube				
*T. aestivum*	Carbon nanotube				
*C. sativa*	PEI-Au@SiO_2_	Transcription factors	Infiltration	Transient expression	[73]
*Z. mays* (calli)	Cationic fluorescence nanoparticle	Protein	Biolistic	Stable transformation	[74]
*N. tabacum* (cotyledons)	Gold capped MSNs	GFP plasmid; chemical expression inducer	Biolistic	Transient expression	[75]
*B. juncea* (hypocotyl explants)	Calcium phosphate NPs (CaPNPs)	*β*-glucuronidase (GUS) plasmid	Passive diffusion	Stable transformation	[76]
*N. tabacum* (protoplasts and leaf explants)	Organically functionalized CNTs	YFP plasmid	Co-culture	Stable transformation	[77]
*A. thaliana* (roots)	Organically functionalized MSNs	mCherry plasmid	Passive diffusion	Transient expression	[78]
*E. sativa, N. benthamiana*, *T. aestivum and G. hirsutum* (leaves)	Polymer-functionalized CNTs	GFP plasmid; siRNA for transgenic silencing	Infiltration	Transient expression	[50]
*S. lycopersicum* (leaves)	Mesoporous silica nanoparticles (MSNs)	*β*-glucuronidase (GUS) plasmid	Spraying and injection	Transient expression	[79]
*C. sativus* (root)	Copper nanoparticles (CuNP)	-	Co-culture	-	[80]
*A. cepa* (epidermis cells)	Gold-plated MSNs	GFP and mCherry plasmids; GFP protein	Biolistic	Transient expression	[81]
*B. napus* L. var. Jet Neuf (protoplasts and walled cell suspension)	Magnetic gold NPs	FITC molecules and *β*-glucuronidase (GUS) plasmid	External magnetic field	Transient expression	[82]
*D. zingiberensis* (calli suspension)	Poly-L-lysine-coated starch NPs	GFP plasmid	Sonoporation	Transient expression	[83]
*A. stolonifera L.* (calli suspension)	Polyamidoamine (PAMAM) dendrimer NPs	GFP plasmid	Co-culture	Transient expression	[84]
*N. tabacum* var. Virginia (root cells)	Single-walled carbon nanotube	GFP plasmid	Co-culture	Transient expression	[85]
*N. tabacum* and *C. purpureus* (protoplasts)	Dimethylaminoethyl methacrylate (DMAEM) polymer NPs	YFP and GFP plasmids	PEG transfection	Stable transformation and transient expression	[86]
*G. hirsutum* (pollen)	Magnetic Fe_3_O_4_ NPs	Selectable marker gene plasmids	External magnetic field	Stable transformation	[46]

## Data Availability

No additional data.

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
