# Peer review of "Application of Nanotechnology in Plant Genetic Engineering"

_ijms, 2023, doi:10.3390/ijms241914836_

Round 1
Reviewer 1 Report
The work by Wu et al. offers a thoughtful and provocative review addressing the perspectives of nanotechnology utilization for plant bio-engineering. The work is well written, and highlights key advances in the field as well as future avenues and promises. The literature compilation is substantial and offers new perspectives to the readers.
However, ss detailed below, my major suggestions are to include a main figure 1 in order to sketch the overall integration of nanotechnology with plant genetic engineering, and to follow a PRISMA procedure (http://www.prisma-statement.org/PRISMAStatement/FlowDiagram) in order to make the review more systematic. Before recommending acceptance, I would also suggest authors to address the following improvements (I was unable to pinpoint line numbers because the document lacks them, please ensure that the following version has line numbers).
First, please list explicit review goals at the end of the introduction section (first paragraph, second page) and a priori hypotheses to be explored by the review. This will allow readers focusing on concrete questions and expected trends on how to effectively utilize nanotechnology within plant genetics.
Second, when reporting literature compilation, I encourage authors to follow PRISMA’s guidelines to prepare reviews. Please include a brief specific methodological section (after the introduction section) explaining the concrete steps/parameters used during the search (i.e. keywords, search equation, target repositories), filtering, and summary of key references. This section can be brief (one paragraph), but yet would make the literature search on nanotechnology for plant genetic engineer more repeatable. I bet carrying out a more systematic search can substantially expand table 1 (for instance, I am enlisting some missing cases in my last comment below).
Third, the review is missing a key summary figure that broadly summarizes the perspective on how to effectively utilize nanotechnology within plant research. I am thinking in a broader figure that generalized current figure 1 and 2, before presenting the two latter. The main figures in Genes 2021 12:783 and Genes 2022 13:1 may serve as guidance.
Fourth, authors missed to integrate modeling tools, such as machine learning, capable to better predict optimized pathways and outcomes of nanotechnology application in plant biology and genetics. When commenting on the utility of machine learning techniques in the context of plant genetic engineer, I invite authors including seminal works that have innovated machine learning techniques to leverage target gene resources (i.e. Nat Rev Genet 2015 16(6):321-32, Trends Genet 2018 34(4):301-12). For instance, convolutional neural network, deep belief network, multivariate Poisson deep learning, multilayer perceptron, probabilistic neural network or radial basis function neural networks may help improving the prediction of target genes for nanotechnology-guided genetic engineer approaches by integrating heterogeneous datasets while side stepping the curse of dimensionality (Front Plant Sci 2020 11:580136). Please couple with my second comment above, specifically by expanding accordingly the keywords and search equation as part of a PRISMA-inspired systematic review.
More specifically, I recognize authors for mentioning thoughtfully throughout the review the goal to achieve tolerance to abiotic stresses (first paragraph in first page). Still, I am missing key examples supporting the utility of candidate genes for genetic engineering approaches aiming to leverage abiotic stress tolerance to heat (e.g. Front Genet 2019 10:954, and Genes 2021 12:556), and drought (e.g. Front Plant Sci 2018 9:128, and PLoS One 2013 8(5):e62898). Ultimately, what are the changes to simultaneously bio-engineer tolerance to various abiotic (and biotic) stresses using nanotechnology?
Last but not least, in order to improve readability, I would recommend splitting the last section (numbered as 5 in the last paragraph of page 8) into individual “Concluding remarks” and “Future Directions” sections. The latter should start by discussing the potential caveats of other studies until now to effectively utilize nanotechnology within plant genetic engineer programs, and propose new avenues of research. Authors must envision novel strategies to assist in the near future further nanotechnology application into the cultivated genepools. Mind the potential of coupling nanotechnology-enhanced genetic engineer with predictive breeding (i.e. genomic selection, e.g. G3 2016 6:1819-1834) and speed breeding. This way the review will have a broader scope, being more interesting to many more readers of IJMS.
Reviewer 2 Report
Plant genetic engineering is an important technology with wide use in producing GM crops and in functional analysis of genes. As the scope of GM crops is limited due to regulatory issues, the current advances in gene editing technology are making significant strides in plant molecular biology. Advances in nanotechnology further enhance the capabilities of plant genetic engineering. In the current review article the authors reviewed the application of nanoparticles in plant genetic engineering. The article is in good shape. Although application of nanoparticles in plant genetic engineering is in very beginning stage, the authors tried to discuss as much possible. However, the article requires some corrections.
1. In para 2 of section 1 (introduction), Line No 6- 8 “In particular, the transit of molecular biology vectors such as DNA, RNA, and proteins to plant cells have become increasingly imperative [5,12].” DNA, RNA and protein are not vectors and they are macromolecules.
2. Check the language for any errors throughout the manuscript. (Eg. Under section 2.1 Line No:4-6, “With the gas pressure, the gold particles carrying DNA through the cell wall, cell membrane, cytoplasm and other layers of structure to reach the nucleus, completing gene transfer. Gene”; DNA pass through)
3. Check the statement “culture (USDA) [32]. In Europe, only two EU countries (Spain and Portug al) cultivated GM crops till 2018 [33].” Few more EU countries also cultivated (Eg. MON810 maize). Make a correction of the statement. Better avoid mentioning number or name of the countries. (Ref: https://doi.org/10.1371/journal.pone.0217272). Moreover, presence of transgene after using any method for genetic engineering will produce GM crops and that requires strict regulation. So better have a separate section to discuss this after discussing all methods.
4. Check the spelling “clay nanosheets, polymer based nanomaterials and mental based nanoparticles, which have been elaborated elsewhere [42,43].” Under section 3.1 line number 2. Mental to metal.
5 Figure 1 looks like it have 2 sections. Need to be clearly divided and explained.
5. Add a column in the Table 1 to mention transient or stable transformation.
Quality of English Language is average. Requires corrections.
Reviewer 3 Report
The current review entitled “Application of Nanotechnology in Plant Genetic Engineering” provides an overview of the progression in plant biomolecular delivery techniques, detailing their characteristics and constraints. Particular emphasis is placed on the forefront of nanotechnology-driven delivery systems. This review explored various nanoparticle types, the preparation of nanomaterials, the mechanisms governing nanoparticle transportation, and their advanced applications in plant genome engineering.
Comments:
The application of nanotechnology in plant genetic engineering holds great promise for improving crop productivity, sustainability, and resilience. However, these are some key considerations need to be discussed within the current version:
1. Nanoparticle uptake and transport: understanding how plants take up and transport nanoparticles is crucial. Factors such as size, shape, and surface chemistry of nanoparticles influence their mobility within plants. Although the current version covered that point, but more details are required with more recent references.
2. A detailed comparison between conventional genetic transformation methods and nanoparticle-mediated biomolecule delivery methods should be incorporated into a table or figure with recent references.
3. Genome editing: nanoscale tools, such as nanoparticles for gene delivery, can improve precision in genome editing. However, ethical concerns, off-target effects, and regulatory compliance are critical considerations need to be illustrated.
4. Biosafety: assessing the environmental and health risks associated with nanomaterials used in agriculture is crucial. More recent references related to that point should be added.
5. Scaling Up: works related to transitioning from laboratory-scale experiments to field-scale applications which presents technical challenges need to be included.
6. Papers which worked on the continuous monitoring and evaluation of nanotechnology applications are essential to be illustrated.
Incorporating these points into the current review, will help ensure that the application of nanotechnology in plant genetic engineering benefits agriculture while minimizing potential risks and challenges.
Minor editing of English language required
Round 2
Reviewer 1 Report
Authors have followed up good improvements. As a couple of last points I would encourage authors to split the last section between and independent perspectives and an independent conclusions section. Also, authors should consider some of the aforementioned suggested work references as per in the previous review round.
Author Response
Dear reviewer, thanks for your suggestions again. We divided the last section into two independent parts ---- “5. Conclusions” and “6. Prospective”. We also added the additional reference you mentioned to these parts.
Reviewer 3 Report
I appreciate the authors' efforts in revising and enhancing their review. The current version suitable for publication following the implemented modifications. Nevertheless, I suggest relocating Figure 3 (entitled "Figure 3: Highlighting future avenues and key issues in the application of nanotechnology") from the main text of the review. The authors may consider including this figure either as supplementary material or as part of a graphical abstract.
Author Response
Dear reviewer, thanks for your advice. We just simply deleted the Fig. 3.